# HEI Efficiency and Quality of Life: Seeding the Pro-Sustainability Efficiency

Eugénia de Matos Pedro [1,*], João Leitão [1,2,3] and Helena Alves [1]

1   Research Center in Business Sciences (NECE), Department of Management and Economics,
    Faculty of Social Sciences and Humanities, University of Beira Interior, 6200-209 Covilhã, Portugal;
    jleitao@ubi.pt (J.L.); halves@ubi.pt (H.A.)
2   CEG-IST—Centre for Management Studies of Instituto Superior Técnico, University of Lisbon,
    1049-001 Lisbon, Portugal
3   ICS—Instituto de Ciências Sociais, University of Lisbon, 1600-189 Lisbon, Portugal
*   Correspondence: eugenia@ubi.pt

**Abstract:** This study assesses the efficiency of higher education institutions (HEIs), considering the social, environmental and cultural factors (pro-sustainability), and at the same time examines how this efficiency can influence regional quality of life (QoL). The study adopts a two-step methodology. In the first step, the standard Data Envelopment Analysis (DEA) is used to estimate the efficiency scores of 23 Portuguese public HEIs; and in the second step, a multivariate logit regression is performed to assess the role played by the HEIs' pro-sustainability efficiency in regional QoL. The main findings reveal that the HEIs located in the Greater Lisbon area have a higher pro-sustainability efficiency, but that efficiency is more significant regarding social factors. Concerning the contribution of pro-sustainability efficiency to the region's QoL, this is significant for all the components, with the environmental and cultural aspects contributing positively to this significance.

**Keywords:** data envelopment analysis; higher education institutions; quality of life; pro-sustainability

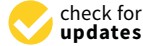

## 1. Introduction

Improved living conditions, decentralised decision-making, well-being and the life expectancy of people and places are increasingly important strategic issues in a world with growing social inequalities and injustice and intensified environmental pressure, as seen in different regions [1]. Work and technological progress are no longer the main factors interfering with economic growth [2], as natural capital and social capital have occupied a prominent place in regional development [3]. A focus only on economic concerns has been re-interpreted as restricted, reductionist and unable to appropriately capture the significant and valuable aspects of individual and social existence, such as health, quality of life (QoL) and well-being [4,5].

The regional availability of knowledge and skills is as important as physical infrastructure, resulting in regionally committed higher education institutions (HEIs) that can become essential and potential assets with a fundamental role in economic [6], social [3] and sustainable [7] development. So, the HEIs' regional involvement, more than a process that can be objectively planned or forecast, is a learning process that characterises the specificities of a subjective deliberation process. This is claimed by [8], according to whom this process is influenced simultaneously by factors operating at the intra-organisational and regional levels, and at the level of the environment the HEIs belong to. Greater HEI involvement is expected with the different agents of the region of influence, incorporating in the former's mission the solid intention to ensure responses to the region's needs and produce improvements also in terms of the resident population's QoL [9]. Therefore, a region's QoL emerges as a multidimensional indicator of performance, which helps to understand the region's situation, as well as the efficiency the HEIs can have in transforming inputs,

arising from national policies, into outputs, with repercussions for the region. HEIs are not only located in places, they belong to them, and so the HEIs' capacities and potential are also shaped by how they interact with their region [9].

Although the function of HEIs has been extended beyond teaching and research, the problem focused by [6] is that often they tend towards internationalization more than regional action, especially with regard to research, reflecting the priorities of governments and their research councils as the those mainly financing that research. It is therefore fundamental to study to what extent HEIs are synchronized with, and contribute to, their region, to understand how efficient HEIs are and whether all their missions are being fulfilled. In addition, social, environmental and cultural impacts stimulate organisations to reconsider their management models, seeking re-dimensioning that goes beyond traditional forms and moves towards pro-sustainability management [10].

Given the importance of resource management nowadays, all organisations are encouraged to have pro-sustainability management, recognise the importance of their social and environmental impacts and carry out actions to reduce their environmental impact [11]. Therefore, pro-sustainability management must also involve social, environmental and cultural variables throughout the process of managing, planning, organising, directing and controlling, using the functions that form that process, as well as the interactions occurring with the region [10]. The HEIs' traditional form of centralized and bureaucratic organisation is now challenged by the need to respond flexibly and pro-sustainably to increasingly unpredictable environmental changes, to become actively involved in the region's various needs and seek sources of finance besides those traditionally associated with teaching and research activities [12]. Furthermore, HEIs are drivers of social and individual development, being endogenous factors of their increased capacities as promoters of human rights such as intellectual solidarity, democracy, peace and justice [13]. The nature of that role will depend on the missions and skills of each HEI, but in all cases, HEIs are the main stimulants in their region in terms of the social and cultural contribution they make to society [14]. In this respect, Boulton and Lucas [15] say that HEIs contribute to regional vitality and serve as agents of social justice and cultural mobility wherever they are located.

Most studies on HEI efficiency focus, above all, on aspects related to teaching and research activities, e.g., [16–18], with a lack of attention paid exclusively to the social, environmental and cultural aspects. For example, Wolszczak-Derlacz [16] identifies as a study limitation the fact of not including variables to measure HEIs' contribution to the surrounding community regarding the social aspects. Since HEIs operate in different environments, studying the transformation of their inputs into outputs related to these environments can bring new contributions and implications able to redefine action strategies, both for HEI managers and regional authorities.

HEIs have several multiplier effects and impacts, the economic ones being the most recognized through several studies, e.g., [19–21]. The approach now operationalized complements the studies focused so far on economic efficiency, incorporating the still unexplored social, environmental and cultural components. Bearing in mind the growing importance of the sustainability of institutions in general, for HEIs in particular, and for the regions in which they are located, this study explores, in a pioneering way, the three components of sustainability, namely, social, environmental and cultural, which contributes to a better understanding and deepening of the HEIs' pro-sustainability orientation. Despite the wide range of previous studies dealing with the economic efficiency of HEIs, a gap was detected in the literature; that is, there is a lack of studies on the different types of HEI efficiency, social, environmental and cultural, jointly treated for assessing the influence of pro-sustainability efficiency, specifically integrating the three types of efficiency variables previously mentioned in terms of determining the regional QoL.

System-wide and transformative change in HEIs are seen as a precondition, which facilitates sustainability [22]. The most efficient HEIs are expected to contribute to the strengthening of regional QoL, since (i) in social terms, they increase the effectiveness of

the use of public money for increasing social cohesion and mobility at the regional level; (ii) in environmental terms, they reduce pollution and waste by educating stakeholders in the region to become more environmentally friendly; and (iii) in cultural terms, they provide a greater access to culture goods and services, and promote different cultural and scientific activities with a high impact.

Considering the context described in the literature of reference, an important contribution of this study will be to assess HEI efficiency with regard to social, environmental and cultural factors, which hereafter will be referred to as pro-sustainability factors, and at the same time indicate the way for HEIs to promote regional QoL, admitting that the latter can be influenced by the efficiency of these institutions. This study adds to the knowledge on HEIs' impact on their regions, at the same time summarising the HEIs' role in transforming society, considering society's pro-sustainable situation. It will also give better orientation to the HEIs' mission of social responsibility, in order to develop reference frameworks considering the external environment, by defining its objectives. It will allow policy-makers and designers of public policies to gain better knowledge of a HEI's potential as an institution rooted in the region and an important actor in present and future development. This is an innovative study that can contribute important knowledge to the literature of reference and to HEIs and their regions.

Considering the above, this study intends to address the following questions:

Q1: Are HEIs efficient in transforming their inputs into pro-sustainability outputs (social, environmental and cultural)?

Q2: What is the role of this efficiency as a predictor of regional QoL?

This research analyses efficiency using the two-step DEA method, to make a comparative analysis of the efficiency of twenty-three Portuguese HEIs. In the first step, the efficiency scores are determined using DEA with different sets of inputs/outputs that have been previously identified through a literature review, and validated through qualitative assessment carried out with diverse HEIs stakeholders. The DEA method and its variants have multiple applications in the literature [23], including in the analysis of the efficiency of HEIs, e.g., [16,23–26].

In the second step, the efficiency scores obtained in the first stage are regressed on a collection of explanatory variables referring to QoL. The regression models commonly used in the second stage include the ordinary least square (OLS), censored regression (e.g., logit, probit and tobit models), truncated regression and panel data models [27]. Banker and Natarajan [28] show that the two-stage approach for the DEA can yield statistically consistent coefficient estimators under certain general distributional assumptions. Johnson and Kuosmanen [29] further show that the estimators remain statistically consistent even when the first-stage input and output variables in the DEA are correlated with the second-stage variables in the regression model [27]. Having in mind the above, in this study, after the efficiency analysis, a logit regression is used, resorting to the multivariate model, to determine the influence of HEI efficiency, using the scores resulting from the DEA analysis, on the QoL of the regions in which the HEIs studied are situated. This type of regression is used to predict categorical placement or the likelihood of category association in a dependent variable based on multiple independent variables. The dimensions used for QoL consider the studies of the OECD (Organisation for Economic Co-operation and Development), Eurostat (Statistical Office of the European Union) and INE (Statistics Portugal).

The article is innovative and contributes to the literature on HEI efficiency in two ways: firstly, it allows mapping of the most efficient HEIs through gathering the key indicators (inputs and outputs) based on studies of the HEIs' impact on regions and from data of a field study, for the purpose of executing a DEA analysis of the pro-sustainability efficiency (social, environmental and cultural); and secondly, it analyses whether this efficiency influences the region's QoL through testing different selected specifications of logit models, of the multivariate type, using the HEIs' efficiency scores as explanatory factors of regional QoL. This type of association, which so far has not been found in the reference literature about sustainable HEIs and regions, increases the knowledge

about HEIs' impact on regions, summarising at the same time the change in the social, environmental and cultural role of the HEIs, considering the population's QoL. It also lets the HEIs strengthen their institutional orientation towards social responsibility and improve their pro-sustainability management, as well as reinforcing the HEIs' role as levers of QoL and social, environmental and cultural sustainability at the regional level.

The paper is structured as follows. The literature on the importance of HEIs for regions and the importance of QoL for regions and HEIs is reviewed, in order to define the inputs and outputs for the purpose of measuring the HEIs' efficiency, as well as the variables to measure the region's QoL. This is followed by a two-stage approach, with presentation and discussion of the results. Finally, the conclusions, implications and limitations of the study are presented.

## 2. Regional Needs and HEIs' Reaction

Activities linked to the new HEIs mission of regional economic development, which involves technology transfer, life-long learning or social involvement [30], are related to the generation, use, application and exploitation of knowledge with all external stakeholders and with society in general, and so this mission cannot be considered as a residual function, but as complementary to the other two missions of teaching and research [31]. A HEI's capacity to respond to the regional needs is influenced by various conditions resulting from the inter-relations between the various geographical levels and the historical legacy of each HEI and its region [6]. HEIs have a great deal to offer, since besides knowledge and human capital, they are crucial drivers of prosperity, inclusion and territorial development, contributing in a wide-ranging way to social questions, environmental innovation and critical reflection, vital in times of challenges and with considerable risks for regions and nations [32]. Geographical proximity and regional involvement are major advantages for HEIs to act as agents of change, promoting human interaction, transferring knowledge and building trust and common purposes among a diversity of actors and interests within regional structures [33].

Considering the unstable external environment, e.g., social and cultural inequalities [34,35] and environmental changes [36], in need of constant innovation, HEIs' behaviour was forced to adopt a strategic business administration, despite the differences between an HEI and a typical business organisation [37,38]. This new situation, in the first phase, led HEIs to draw up innovative competitive strategies with the triple purpose of attracting, capturing and retaining students, ensuring or increasing their participation in the market [39]. However, more recently, HEIs have been adopting new strategies more focused on fulfilling their third mission, directed towards the transfer of knowledge and technology, life-long learning and social, environmental and cultural responsibility [30,40]. Despite their different missions and histories, most HEIs consider the social, environmental and cultural contribution as part of their role, as they contribute to the regeneration of urban and rural areas, social services and health, library services, research to benefit the community and cultural and environmental development [41], among other domains. More than involving active academic participation to create economic, social and environmental programs that improve living standards, generate empowerment and respect interdependence [42], this means that sustainability must go further than acquiring knowledge on issues related to sustainability to provide a transformation of the dominant ways of being and understanding this new social reality [43]. Knowing regional asymmetries can provide educational systems with opportunities to find innovative solutions in disadvantaged areas [44]. Mainardes et al. [38] argue that organisations adapting a strategy to their external environment is a principle of competitiveness. They also say that the Theory of Territorial Competitiveness [45,46] is framed in the current conjuncture of HEIs' competitive management. Consequently, considering this theory, strategies for competitiveness include an important component, local territorial aspects [47]. The strategic path of organisations is to draw up their structures and operations to be linked to their territory of action; i.e., the place where they act defines how these organisations work [45]. In this

regard, Mainardes et al. [38] underline that it is the local community and its actors who define and seek an integrated development strategy, in shared pursuit of solutions to their problems. In this context, standing out are local HEIs, which assume the strategy of market competition, considering companies' needs and preparing professionals who will act in these local companies, thereby creating economic advantages [38]. At the same time, they contribute to a healthier environment where social, cultural and environmental disparities are less pronounced. Therefore, the economic and social benefits obtained by the HEIs are considerable, making them more competitive in the education market and meaning they fulfil their mission in society [45,46].

Contrasting with the first two pillars of HEIs (research and teaching), regional involvement is a multifaceted phenomenon and difficult to delimit [8]. A wide-ranging study involving 14 countries in five continents and carried out by the OECD [48] found that the joint development trajectories of HEIs and their respective regions are shaped by the combination of a wide range of factors that influence and are influenced. This study, which draws attention to and "begins a debate" on the importance of HEIs for their regions of influence, mentions that few take the surrounding environment into consideration. Taking the example of a study by Radinger-Peer [8] in the region of Kaiserslautern (Germany), the multi-level environment in which the HEI is situated influences the HEIs' regional involvement. This shows that the occurrence of activities with a regional commitment (commercial and non-commercial) cannot be explained only by individual indicators (e.g., gender, age and experience), but must be accompanied more systematically and interactively.

## 3. HEIs' Efficiency in Their Region of Influence

A critical factor of HEI positioning over time has to do with the nature, number and distribution of organisations in a given place, which depends on the availability of resources and the level of competition, making environments competitive [49,50]. In the light of Neo-Institutional Theory, HEIs' positioning is generated by the search for legitimacy to deal with the external pressures of their surrounding environment [49]. Consequently, in the positioning process, only the capacity to differentiate from competitors, through creating a unique profile that cannot be reproduced, lets HEIs obtain competitive advantages [51,52]. Olivares and Wetzel [53] say that a unique position is built through inputs (combination of resources used) and outputs (activities provided) and effective and efficient processes, with implications for the capacity to implement and manage the most suitable combinations of an inputs–outputs process [54]. Therefore, HEIs' increased strategic planning capacity makes them more efficient with their resources and become more pro-active in anticipating changes and in developing the capacity to respond suitably to the needs encountered [51].

Oliver's theory states that the capacity to respond to organisational pressure or political objectives is delimited by legitimacy or social efficiency [55]. Legitimacy is a subjective interpretation found in the beliefs and perceptions of individuals and groups in relation to the actions and behaviour of others [56]. Efficiency is essentially the comparison between the inputs used in certain activities and the outputs produced [57]. In the case of HEI efficiency, it refers to a comparison of the marginal social costs and benefits, and does not solely relate to a comparison of the HEIs' costs and revenues [58]. Therefore, legitimacy and efficiency are two concepts that should not be disassociated when speaking about the HEIs' influence on their regions: firstly, because legitimacy implies a general trust in society that a given entity's power to make binding decisions is justified and appropriate [59]; and secondly, because the HEI's efficiency has repercussions on the region through economic impacts arising from public investment, spending on general consumption, jobs created and, in particular, students' spending in the region, as well as that of the academic community in general. In turn, this expenditure has an impact on regional indicators: (i) through the volume of business, employment, income, property values and local authority expenditure [60]; (ii) through impacts caused by the indirect supply of services such as health, sport, culture, technology transfer and others; and (iii) through the HEIs' transformational activities arising from the improved quality of the local economies

and political systems [61] and the supply of services that serve as inputs to the region. These transformational activities are legitimized by the processes through which the HEIs organise their fundamental tasks, such as teaching, research and activities arising from the third mission, through transforming their inputs into outputs for the region.

*Definition of the Key Indicators to Measure Efficiency*

There is extensive literature on measuring HEIs' efficiency [18,62]. In HEIs, efficiency can be measured though various techniques and approaches, considering the subject of analysis and the characteristics of the organisations to be studied [63]. Despite differences in the methods used (i.e., parametric and non-parametric) and in the details of model specification, all existing studies consider higher education activity as combining inputs (e.g., human and financial resources, premises, etc.) to produce important outputs (results) such as education (e.g., number of graduates), research (publications), knowledge transfer (patents, academic spin-offs, public events, etc.) [64] and social, cultural and environmental involvement (e.g., cultural and social activities) [65]. The inputs and outputs vary substantially from one study to another, making it necessary to make a survey to have a general idea of the most commonly used key indicators. As this study focuses on assessing pro-sustainability efficiency, these inputs and outputs will be seen as part of the HEIs' social, cultural and environmental objectives.

To determine the key indicators (inputs-outputs) used to measure HEI efficiency, from the perspective of the effects of those institutions on the region, various previous studies were first considered [19–21,65–69]. It is worthy of note that some of the indicators that were identified in the scope of the current study are partially influenced by the HEIs, since the former were collected at the regional level. Similar variables were also used in the previous studies included in this literature review; for example [69] stated that the outputs of HEIs can be measured through their impacts on the social and environmental well-being of the region.

Then, to identify and validate the key indicators found in those studies, a semi-structured interview script was drawn up. These interviews were held face-to-face or via Skype with 20 relevant individuals in the academic, political and social spheres, as well as with the economic agents resident in regions in which the HEIs are located. Next, the results of the interviews were discussed and analysed in a meeting of the research group.

Given the need to identify the HEIs' inputs in order to measure these institutions' impact on their surrounding region, taking as reference the studies of Goldstein and Renault [21], Jonkers et al. [68] and Skyrme and Thompson [69], a framework of analysis was defined to classify the inputs proposed by the interviewees in seven categories: the HEI's Economic Support (income); the HEI's expenditure (expenses); the HEI's students; the Employment in the HEI and provision of qualified work; the volume of service provision activities; the HEI's institutions/R&D centres; and the HEI's social and cultural environment (see Table 1).

Similarly to the procedure described above, and given the need to identify the HEIs' outputs able to measure these institutions' impact on their surrounding region, considering the social, civic and environmental outputs in the local and regional surroundings, as referred by Drucker and Goldstein [70], Kroll and Schubert [71] and Skyrme and Thompson [69], a framework of analysis was defined for classifying the results/indicators proposed by the interviewees, in terms of social, environmental and cultural factors (pro-sustainability), as advocated by Alves et al. [10] (see Table 1).

For the purpose of the efficiency analysis (DEA), the key inputs/outputs and respective indicators presented in Table 1 were defined, based on the year 2018. Three aspects were considered: interviewees' classification of the variables proposed in the interview script; interviewees' answers to the open questions; and the availability of data for collection. The indicators were gathered on the platforms of INE, PORDATA (Database of Contemporary Portugal) and the Sales Index (Marktest Group) database, according to data available at the NUTS III level. Data referring to the HEIs were gathered from the institutions'

activity reports, management and accounting reports and websites. Regarding the students' expenditure, the figures were gathered from the study made by [72]. Table 1 presents the key indicators determined.

**Table 1.** Inputs and outputs: Key indicators to measure pro-sustainability efficiency.

| INPUTS/Indicators | OUTPUTS/Indicators |
|---|---|
| HEI's Economic Support (income)<br>I1—Ratio: Own income/SB<br>HEI's Expenditure (expenses)<br>I2—Ratio: Expenditure on staff/SB<br>HEI's Students<br>I3—Ratio: No. of 1st-cycle students */total students<br>Employment in the HEI and Provision of Qualified Work<br>I4—Ratio: Total no. of lecturers and researchers/total students<br>Volume of Service Provision Activities<br>I5—Ratio: Amount declared in service provision/total own income<br>HEI's Institutions/R&D Centres<br>I6—Ratio: No. of publications ISI/total no. of publications (ISI + SCOPUS)<br>HEI's Social and Cultural Environment<br>I7—Rate of scientific, cultural and social, and sporting events<br>I8—Ratio: Student's annual cost of living (per HEI)/national minimum salary | Social Pro-Sustainability<br>O1A—Ratio: Total no. of social action grants awarded/total grants requested<br>O1B—Access to broadband internet per 100 inhabitants (%)<br>O1C—Proportion of women in higher education graduates<br>O1D—Inequality in the distribution of the declared gross income of tax aggregates<br>Environmental Pro-Sustainability<br>O2A—Wastewater treatment stations (No.)<br>O2B—Municipal expenditure on the environment per capita: by management domains and environmental protection<br>O2C—Environmental invention patents registered by HEIs and research institutions per region (No.)<br>O2D—Investment in protecting municipal biodiversity and landscape<br>Cultural Pro-Sustainability<br>O3A—Municipal expenditure on cultural and creative activities (€)<br>O3B—No. of people in cultural and social, and sporting activities<br>O3C—Cultural premises/facilities (No.)<br>O3D—Municipal expenditure on sporting activities and equipment (€) |

\* Polytechnic education includes the variant of Professional Technical Course. Legend: SB: State Budget; ISI: International Scientific Indexing Web of Science; SCOPUS: SciVerse Scopus. Source: Own elaboration.

## 4. Regional Quality of Life and HEIs

Regional quality of life reflects the levels of regional disparities within different countries, separating privileged and lagging regions with respect to standards of living and individual well-being [73]. Although economic factors are important in determining the attractiveness of regions for organisations, the quality of the environment (social, political, natural, etc.) also plays an important role [74]. In order to create economic growth, it is essential to strengthen competitiveness, and an important aspect of this competitiveness is QoL [74].

Briefly, regions wish to attain a balance of economic, social, environmental and cultural standards, so that the resident population can enjoy an excellent QoL. It is assumed that HEIs can contribute to that improvement and increase the QoL through research done on the university campus and the transfer of knowledge to society, and by providing the surrounding area with a wide variety of cultural, sporting and social activities [75], increasing the education, qualifications and mobility of human capital [76–78], developing and raising technological levels, increasing productivity and consequently improving the region's economic performance [76,77]. HEIs are therefore expected to be able to contribute to a region's attractiveness and development, as drivers of positive externalities regarding QoL [79].

As suggested by Shapiro [80], the local human capital level increases the implicit value of an area's consumption amenities; i.e., the stock of human capital transforms an area into a more desirable place to live and increases the QoL [81]. From this line of reasoning, it is reasonable to assume that the QoL of the region surrounding the HEI can be influenced positively or negatively by this type of institution, both through the human capital that carries out its activities and if a good QoL is ensured within the institution; this

can be reflected positively (1) through the human capital's behaviour, in the environment surrounding the institution and through the surrounding systems that support the HEIs' human capital and the population in general, through creating better living conditions, more employment, better health services, better artistic and cultural services, more areas for recreation and leisure, etc., causing them to feel satisfied and content, and (2) through the HEI itself, which should match its objectives, missions and values to the needs encountered. The following sub-section defines the dimensions for measuring regional QoL.

*Dimensions for Measuring Regional QoL: Definition*

Recently, the OECD has become deeply involved in the debate on the most appropriate way to measure a population's well-being and has made studies in the area of QoL. The OECD created the Better Life Index (In http://www.oecdbetterlifeindex.org), with 11 variables reflecting well-being in terms of the material conditions of life: housing, income and work; and in terms of QoL: community, education, environment, governance, health, satisfaction with life, safety and work–life balance.

The European Union, through Eurostat, also divulged through its online publication (In http://ec.europa.eu) an index designated as Quality of Life Indicators (measuring the quality of life). This index presents nine dimensions of QoL: material living conditions, production or main activity, health, education, leisure and social interactions, economic and physical security, governance and basic rights, natural, living environment and general experience of life.

INE [82] presented an index of 10 indicators of well-being in two dimensions. The first, referring to the material living conditions, includes economic well-being, economic vulnerability and work and remuneration. The second, referring to QoL, covers health, work–life balance, education, knowledge and competences, social relations and subjective well-being, civic participation and governance and personal and environmental safety.

Accordingly, and using the indicators available, for the year 2018, on the INE, POR-DATA and Sales Index platforms, for NUTS III (Nomenclature of Territorial Units for Statistics), to measure the QoL the following composite index is considered: material life conditions, health, education, environment and leisure and safety. It is noted that these indicators are common in the three examples presented (OECD, Eurostat and INE). Therefore, for the purpose of measuring regional QoL, Table 2 presents the dimensions, respective measurement indicators and codes attributed.

**Table 2.** Data to measure regional quality of life (QoL).

| Dimension/ Variable | Indicators (NUTS III) | Codes |
|---|---|---|
| Material life conditions | - Credit granted to customers by banks, savings banks and mutual agricultural banks | Credit |
| | - Unemployment registered per 100 inhabitants aged 15 or older | Unemployment |
| | - Purchasing power per capita | Purchasing power Housing |
| | - Housing loan per inhabitant | |
| Health | - No. state hospital beds universally available and in hospitals in public-private partnership by geographical location | Hospitals Deaths Health activities |
| | - Deaths of residents in Portugal from certain causes | |
| | - Average No. of people working in human health and social support activities | |
| Education | - Gross rate of schooling in higher education | Schooling. Higher Ed. Non-Higher Ed. Estab. Higher Ed. Estab. Computers |
| | - No. of non-higher education establishments | |
| | - No. of higher education establishments | |
| | - No. of computers in primary and secondary education | |

| Dimension/ Variable | Indicators (NUTS III) | Codes |
|---|---|---|
| Environment | - Ratio: Municipal expenditure on the environment per capita (environmental management and protection) | Environment Waste NGEO |
| | - Separated urban waste collected per inhabitant | |
| | - Non-governmental environmental organisations (NGEO): number | |
| Leisure | - No. of cultural premises/facilities | Premises Artistic activities Museums |
| | - No. of people in artistic, performance, sporting and recreational activities - No. of museums | |
| Safety | - Crime rate (‰) - Crimes registered by police - No. of inhabitants per firefighter - No. of accidents | Criminality Crimes Firefighter Accidents |

Source: Own elaboration.

## 5. Methodological Design

This methodology includes two-stages. In the first stage, the DEA was used to measure the HEIs' efficiency, considering the inputs and outputs presented in Table 1 (p. 6); the Frontier Analyst Application (version 4.4.0) was used to execute the DEA. In the second stage, a multivariate logit regression was used, with the scores generated in the DEA analysis and from the dimensions of QoL presented in Table 2 (p. 8); in this analysis, Stata software version 15.1 was used.

The study aims to determine the efficiency of 23 state HEIs in Portugal, using data referring to 2018. The data were obtained from the INE, PORDATA and Sales Index databases, also including elements available such as their activity reports, management and accounting reports, as well as information on the websites of the HEIs analysed.

Selection of the 23 HEIs for this study was according to the following criteria: (i) Portuguese state HEIs, universities and polytechnics; (ii) state universities belonging to the Council of Rectors of Portuguese Universities (CRUP), an entity coordinating university teaching in Portugal; (iii) HEIs located in each of the 7 regions of Portugal at the NUTS II level, since Portugal has an asymmetric socio-economic situation between regions [83]; and (iv) complete data availability for the year 2018.

### 5.1. DEA Analysis

The DEA model evaluates efficiency by forming performance measures obtained as ratios of the multiple inputs and multiple outputs selected in Table 1. The DEA method was first developed by Charnes et al. [84], who proposed the CCR model (initials of Charnes, Cooper and Rhodes), also known as the CRS model (Constant Returns to Scale). This model, which establishes an analysis with constant returns to scale, determines a proportional relationship between the inputs and outputs, similar to a regression. Years later, the BCC model (initials of Banker, Charnes and Cooper) appeared, also known as VRS (Variable Returns to Scale), proposed by Banker et al. [85], which considers the variable returns to scale.

In this case, the relation between the inputs and outputs is not linear, but convex. According to the same authors, aiming to ensure maximum efficiency, these two basic DEA models can be designed in two ways: (i) oriented towards the inputs: so as to minimize the inputs allocated, maintaining the level of outputs; and (ii) oriented towards the outputs: so as to maximize the outputs, maintaining the level of inputs. Following Agasisti [86], this study uses the CCR type of model, which is an output-oriented framework, because it wants to establish an analysis with constant returns to scale, as well as to determine

a proportional relationship between the inputs and outputs, where the inputs are fixed. The aim is to maximize the outputs; i.e., the outputs directly reflect the input levels.

Before starting the efficiency analysis, it is always useful to have an idea of the data we are going to deal with [87], and so descriptive statistics of the inputs and outputs presented in Table 1 is offered in Table 3.

**Table 3.** Descriptive statistics of the higher education institutions' (HEIs') inputs and outputs (*n* = 23).

| Variable | Mean | Stand. Dev. | Min Value | Max Value |
|----------|------|-------------|-----------|-----------|
| I1 | 0.558 | 0.327 | 1.02 | 1.65 |
| I2 | 1.261 | 0.134 | 0.31 | 0.92 |
| I3 | 0.676 | 0.211 | 0.03 | 0.12 |
| I4 | 0.081 | 0.019 | 0.00 | 0.78 |
| I5 | 0.147 | 0.168 | 0.07 | 0.72 |
| I6 | 0.481 | 0.140 | 0.00 | 1.73 |
| I7 | 0.314 | 0.442 | 2.35 | 3.24 |
| I8 | 2.799 | 0.243 | 0.00 | 0.84 |
| O1A | 0.735 | 0.164 | 0.02 | 1.21 |
| O1B | 0.287 | 0.500 | 0.95 | 1.16 |
| O1C | 1.015 | 0.054 | 0.86 | 1.17 |
| O1D | 0.999 | 0.103 | 0.00 | 0.18 |
| O2A | 0.054 | 0.051 | 0.41 | 1.68 |
| O2B | 1.016 | 0.326 | 0.00 | 0.36 |
| O2C | 0.186 | 0.136 | 0.00 | 0.14 |
| O2D | 0.063 | 0.051 | 0.02 | 0.23 |
| O3A | 0.087 | 0.090 | 0.00 | 0.47 |
| O3B | 0.129 | 0.188 | 0.02 | 0.15 |
| O3C | 0.072 | 0.047 | 0.02 | 0.16 |
| O3D | 0.067 | 0.052 | 1.02 | 1.65 |

According to Mainardes [88], it is necessary to find a point of balance in the number of DMUs and indicators, with a view to extend the discriminatory power of the DEA, which can require the insertion or exclusion of indicators during the analysis process. The validity of the DEA should be confirmed through a decision rule formulated by Avkiran [89], according to which the ratio between the number of DMUs and the product between the number of inputs and outputs must be above 1.333 (e.g., No. DMU/(No. inputs * No. outputs). If this rule is not respected, there will be the possibility of a large number of DMUs positioning on the frontier established, which contributes to reducing the DEA's capacity to make a valid discrimination between efficient and inefficient DMUs [88].

For each model, we consider two inputs and one output active for the purpose of output maximization. With this more stratified type of analysis, it is possible to find DMUs that stand out at specific points, which would not happen if the efficiency analysis was general [88]; i.e., if it included all the variables studied simultaneously. This stratification also reveals which variables are most important and those needing greater attention. Accordingly, considering the indicators presented in Table 1, it was necessary to create a composite indicator (CI) for each factor forming the pro-sustainability efficiency, namely, social, environmental and cultural, and a CI joining these three factors in a single output.

Following Daraio and Simar [87], there are several multivariate statistical tools that may be of interest to see a multivariate dataset, e.g., [90]. One of the most-known tools is the normalized principal component analysis (PCA). This kind of analysis aims at reducing the information contained in a multivariate space, providing illustrations in two dimensions. Firstly, it can analyse the correlation structure existing among the variables, and secondly, all the individuals are projected on a reduced two-dimensional space [87]. The observation of the correlation matrix reported in Table 4 tells us that the correlation among all the inputs and outputs, in most cases, is not a problem. To complement this information, see Table 4, in which the correlations of the first two principal components with the original variables are reported. It appears that the information is quite homogeneous among all the variables. This is the information provided by the cumulated percentage of variance explained by the eigenvalues reported in Table 5. In sequence, the correlations of the first two principal components (PCs) with the original variables are displayed below in Table 6.

**Table 4.** Correlation matrix of the inputs and outputs of the HEIs (*n* = 23).

| Variable | I1 | I2 | I3 | I4 | I5 | I6 | I7 | I8 | O1A | O1B | O1C | O1D | O2A | O2B | O2C | O2D | O3A | O3B | O3C | O3D |
|---|---|---|---|---|---|---|---|---|---|---|---|---|---|---|---|---|---|---|---|---|
| I1 | 1 | | | | | | | | | | | | | | | | | | | |
| I2 | 0.461 * | 1 | | | | | | | | | | | | | | | | | | |
| I3 | −0.217 | −0.329 | 1 | | | | | | | | | | | | | | | | | |
| I4 | −0.187 | −0.087 | 0.017 | 1 | | | | | | | | | | | | | | | | |
| I5 | 0.144 | −0.011 | 0.065 | 0.317 | 1 | | | | | | | | | | | | | | | |
| I6 | 0.193 | −0.369 | −0.260 | −0.097 | −0.344 | 1 | | | | | | | | | | | | | | |
| I7 | 0.181 | 0.219 | −0.613 ** | 0.098 | −0.136 | 0.055 | 1 | | | | | | | | | | | | | |
| I8 | 0.108 | 0.267 | −0.053 | −0.193 | 0.123 | −0.376 | −0.058 | 1 | | | | | | | | | | | | |
| O1A | 0.030 | 0.107 | −0.137 | 0.519 * | 0.229 | 0.105 | −0.047 | 0.142 | 1 | | | | | | | | | | | |
| O1B | 0.088 | 0.147 | −0.240 | −0.246 | −0.184 | 0.154 | −0.052 | 0.056 | −0.416 * | 1 | | | | | | | | | | |
| O1C | −0.307 | −0.353 | 0.194 | 0.159 | 0.342 | −0.022 | −0.170 | −0.401 | 0.175 | −0.388 | 1 | | | | | | | | | |
| O1D | 0.051 | 0.047 | −0.244 | −0.256 | −0.123 | 0.366 | −0.086 | −0.071 | −0.344 | 0.882 ** | −0.328 | 1 | | | | | | | | |
| O2A | −0.211 | −0.196 | 0.138 | 0.054 | −0.114 | 0.091 | −0.115 | −0.461 * | −0.088 | −0.002 | 0.253 | −0.103 | 1 | | | | | | | |
| O2B | −0.336 | −0.149 | 0.080 | 0.074 | −0.206 | 0.095 | −0.118 | −0.420 * | −0.189 | 0.297 | 0.297 | 0.283 | 0.094 | 1 | | | | | | |
| O2C | 0.083 | 0.233 | −0.222 | −0.102 | 0.018 | −0.184 | 0.041 | 0.229 | −0.326 | 0.706 ** | −0.332 | 0.594 ** | 0.112 | 0.190 | 1 | | | | | |
| O2D | −0.060 | 0.072 | −0.122 | −0.083 | −0.169 | 0.075 | −0.188 | 0.072 | −0.308 | 0.668 ** | −0.399 | 0.647 ** | −0.174 | 0.566 ** | 0.531 ** | 1 | | | | |
| O3A | 0.351 | 0.155 | −0.343 | −0.154 | −0.031 | 0.275 | 0.099 | −0.126 | −0.395 | 0.878 ** | −0.387 | 0.858 ** | −0.083 | 0.212 | 0.588 ** | 0.611 ** | 1 | | | |
| O3B | 0.220 | 0.161 | −0.293 | −0.210 | −0.110 | 0.214 | 0.018 | −0.027 | −0.420 * | 0.870 ** | −0.407 | 0.896 ** | −0.047 | 0.273 | 0.662 ** | 0.670 ** | 0.866 ** | 1 | | |
| O3C | 0.159 | 0.024 | −0.330 | −0.180 | −0.135 | 0.399 | −0.048 | −0.126 | −0.326 | 0.841 ** | −0.276 | 0.862 ** | −0.026 | 0.272 | 0.482 * | 0.660 ** | 0.875 ** | 0.873 ** | 1 | |
| O3D | 0.560 ** | 0.198 | −0.344 | −0.211 | 0.012 | 0.316 | 0.150 | −0.049 | −0.310 | 0.645 ** | −0.432 * | 0.686 ** | −0.219 | 0.008 | 0.420 * | 0.459 * | 0.898 ** | 0.796 ** | 0.689 ** | 1 |
| Kurtosis | 2.005 | 1.271 | −0.449 | −0.59 | 2.755 | −1.223 | 2.115 | 0.03 | −4.431 | 1.468 | 1.086 | 0.705 | 1.527 | 0.008 | 0.01 | 0.106 | 0.948 | 1.276 | 0.482 | 0.607 |
| Sweetness | 5.685 | 2.402 | −1.409 | 0.846 | 9.129 | 3.213 | 4.332 | −0.842 | 2.629 | 0.16 | 0.558 | −0.836 | 1.867 | −0.102 | −1.572 | −1.662 | −1.055 | −0.202 | −1.063 | −1.335 |
| VIF | 2.402 | 2.929 | 2.667 | 1.339 | 1.912 | 3.327 | 2.031 | 1.359 | 9.910 | 2.700 | 6.768 | 1.618 | 3.653 | 4.783 | 7.981 | 3.154 | 8.399 | 9.486 | 6.274 | 1.271 |

* The correlation is significant at the 0.05 level (2 tails). ** The correlation is significant at the 0.01 level (2 tails).

**Table 5.** Eigenvalues and percentage of variance explained by the HEIs' inputs and outputs (*n* = 23).

| Eigenvalues | % of Variance | Cumulated % |
|---|---|---|
| 6.984 | 0.349 | 0.349 |
| 2.802 | 0.140 | 0.489 |
| 1.969 | 0.098 | 0.588 |
| 1.592 | 0.080 | 0.667 |
| 1.367 | 0.068 | 0.736 |
| 1.221 | 0.061 | 0.797 |
| 0.928 | 0.046 | 0.843 |
| 0.865 | 0.043 | 0.886 |
| 0.727 | 0.036 | 0.923 |
| 0.432 | 0.022 | 0.944 |
| 0.327 | 0.016 | 0.961 |
| 0.214 | 0.011 | 0.971 |
| 0.171 | 0.009 | 0.980 |
| 0.160 | 0.008 | 0.988 |
| 0.101 | 0.005 | 0.993 |
| 0.089 | 0.005 | 0.997 |
| 0.040 | 0.002 | 0.999 |
| 0.009 | 0.001 | 1.000 |
| 0.002 | 0.000 | 1.000 |
| 0.000 | 0.000 | 1.000 |

**Table 6.** Correlations of the first two principal components (PCs) with the original variables (factors loadings) of the HEIs' inputs and outputs (*n* = 23).

| Original Variable | First PC | Second PC |
|---|---|---|
| I1 | 0.103 | −0.363 |
| I2 | 0.074 | −0.391 |
| I3 | −0.136 | 0.241 |
| I4 | −0.109 | 0.043 |
| I5 | −0.072 | −0.110 |
| I6 | 0.104 | 0.159 |
| I7 | 0.023 | −0.242 |
| I8 | −0.001 | −0.376 |
| O1A | −0.170 | −0.142 |
| O1B | 0.352 | 0.065 |
| O1C | −0.185 | 0.301 |
| O1D | 0.343 | 0.103 |
| O2A | −0.041 | 0.295 |
| O2B | 0.102 | 0.399 |
| O2C | 0.256 | −0.036 |
| O2D | 0.277 | 0.125 |
| O3A | 0.362 | −0.003 |
| O3B | 0.368 | 0.032 |
| O3C | 0.338 | 0.104 |
| O3D | 0.312 | −0.133 |

Having in mind these results, to build the CIs, the percentage corresponding to the information of each variable in the initial model was added to the initial value of each indicator, which will constitute the social output, the environmental output, the environmental output and the pro-sustainability output.

The formulas used in building the CIs were the following:

CI 1: Social output (SO): [((O1A × 0.036) + (O1B × 0.022) + (O1C × 0.016) + (O1D × 0.011))/4]     (1)

CI 2: Environmental output (EO): [((O2A × 0.009) + (O2B × 0.008) + (O2C × 0.005) + (O2D × 0.005))/4]     (2)

CI 3: Cultural output (CO): [((O3A × 0.002) + (O3B × 0.001) + (O3C × 0.000) + (O3D × 0.000))/4]     (3)

CI 4: Pro-sustainability output (PSO): [((O1A × 0.036) + (O1B × 0.022) + (O1C × 0.016) + (O1D × 0.11)

+(O2A × 0.009) + (O2B × 0.008) + (O2C × 0.005) + (O2D × 0.005) + (O3A0.002) + (O3B × 0.001)    (4)

+ (O3C × 0.000)+ (O3D × 0.000))/12)]

where O1A = ratio: total no. of social action grants awarded/total grants requested; O1B = % access to broadband internet per 100 inhabitants; O1C = proportion of women among higher education graduates; O1D = inequality in the distribution of declared gross income of households for tax purposes; O2A = no. of waste water treatment stations; O2B = municipal expenditure in the areas of environmental management and protection; O2C = no. of environmental invention patents registered by the HEIs and research institutions; O2D = investment in protecting biodiversity and the municipal landscape; O3A = municipal expenditure on cultural and creative activities; O3B = no. of people in cultural, social and sporting activities; O3C = no. of cultural premises/facilities; and O3D = municipal expenditure on sporting activities and equipment.

Table 7 presents the models defined for the DEA analysis.

**Table 7.** The DEA models.

| Model No. | Input (I) | Outputs (O) | | | |
| --- | --- | --- | --- | --- | --- |
| | | Social Models (A) | Environmental Models (B) | Cultural Models (C) | Pro-Sustainability Models (D) |
| 1 | I1—Ratio: own income/SB<br>I2—Ratio: expenditure on staff/SB | Social output (SO) | Environmental output (EO) | Cultural output (CO) | Pro-sustainability output (PSO) |
| 2 | I3—Ratio: no. of 1st-cycle students/total students<br>I4—Ratio: total no. of lecturers and researchers/total students | | | | |
| 3 | I5—Ratio: declared value of service provision/total own income<br>I6—Ratio: no. de publications ISI/total no. of publications (ISI + SCOPUS) | | | | |
| 4 | I7—Rate of scientific, cultural, social and sporting events<br>I8—Ratio: student's annual cost of living (by HEI)/national minimum salary | | | | |

Source: Own elaboration.

### 5.2. Multinomial Logit Model Analysis

The second stage assesses the technical efficiency, through a multinomial logit regression analysis. Logit regression was chosen because it is a regression technique that is used to model the occurrence, in probabilistic terms, of one of the two achievements of the classes of the dependent variable, where the independent variables can be qualitative or quantitative; the logistic model allows to evaluate also the significance of each of the independent variables in the model [91]. The multinomial logit model was chosen because is used to predict categorical placement or the likelihood of category association in a dependent variable based on multiple independent variables, with independent variables being either dichotomous (i.e., binary) or continuous (i.e., interval or proportion in scale) [92]. As in the binary logistic regression, multinomial logistic regression uses the maximum likelihood estimate to assess the likelihood of categorical association [92]. The data were analysed using SPSS software (vs 25). The dependent variables are presented in Table 2: material living conditions, health, education, environment, leisure and safety. The independent variable is HEI efficiency, which was calculated from the scores produced by the CCR models.

First, all the values of the variables were normalized and then the dependent variable (QoL) and independent variables (social efficiency, environmental efficiency, cultural efficiency and pro-sustainability efficiency) were transformed in polychotomous nominal variables, presenting three mutually exclusive classes. In order to identify the intervals, namely, 0 (weak variation), 1 (average variation) and 2 (high variation), an algorithm was

used: (i) the maximum and minimum variation was found in each of the variables; (ii) the maximum (M) minus the minimum (m) to be divided by two was calculated to find the size of each interval (s); and (iii) three intervals were built incrementally: [m,m + s[;[m + s,m + s+s[;[m + s+s,M[. Then, a final dummy variable was introduced, for control purposes, aiming to determine whether the HEI's size, according to the number of students enrolled, had a significant effect on the results. Two regression models will be considered: Model 1, including the independent variables "social efficiency", "environmental efficiency", "cultural efficiency" and the control variable "size"; and Model 2, considering as the independent variable "pro-sustainability efficiency" and as the control variable "size". Both models have the dependent variable of QoL. Table 8 presents the variables included in this study as well as the measurement scales defined.

**Table 8.** Variables of the multinomial logistic regression and measurement scales.

| Model | Type | Description | Scales/Measurement |
|---|---|---|---|
| 1 and 2 | Dependent | QoL (life conditions + health + education + environment + leisure + security) | [≥0.370 and >0.637[=0; [≥0.637 and <0.903[=1; [≥0.903 and <1.170[=2 |
| 1 | Independent | SE: Social efficiency scores obtained from the DEA analysis | [≥−1.436 and >−0.375[=0; [≥−0.375 and <0.686[=1; [≥0.686 and<1.747[=2 |
| 1 | Independent | EE: Environmental efficiency scores obtained from the DEA analysis | [≥−1.737 and >−0.548[=0; [≥−0.548 and <0.641[=1; [≥0.641 and <1.829[=2 |
| 1 | Independent | CE: Cultural efficiency scores obtained from the DEA analysis | [≥−0.762 and >0.203[=0; [≥0.203 and <1.167[=1; [≥1.167 and <2.132[=2 |
| 2 | Independent | PS: Pro-sustainability efficiency scores obtained from the DEA analysis | [≥−1.476 and >−0.425[=0; [≥−0.425 and <0.626[=1; [≥0.626 and <1.676[=2 |
| 1 and 2 | Independent | SIZE-Size by n° of students in the HEI | = 0 < Average value of the n° of students enrolled = 1 ≥ Average value of the n° of students enrolled |

Source: Own elaboration.

In general terms, the multinomial logistic regression model estimator is represented by the following:

$$P(Y=0|X) = \frac{e^{\beta_{00}+\beta_{01}X_1+...+\beta_{0p}X_p}}{e^{\beta_{00}+\beta_{01}X_1+...+\beta_{0p}X_p}+\ e^{\beta_{10}+\beta_{11}X_1+...+\beta_{1p}X_p}+\ e^{\beta_{21}+\beta_{01}X_1+...+\beta_{2p}X_p}} \quad (5)$$

$$P(Y=1|X) = \frac{e^{\beta_{10}+\beta_{11}X_1+...+\beta_{1p}X_p}}{e^{\beta_{00}+\beta_{01}X_1+...+\beta_{0p}X_p}+\ e^{\beta_{10}+\beta_{11}X_1+...+\beta_{1p}X_p}+\ e^{\beta_{21}+\beta_{01}X_1+...+\beta_{2p}X_p}} \quad (6)$$

$$P(Y=2|X) = \frac{e^{\beta_{20}+\beta_{21}X_1+...+\beta_{2p}X_p}}{e^{\beta_{00}+\beta_{01}X_1+...+\beta_{0p}X_p}+\ e^{\beta_{10}+\beta_{11}X_1+...+\beta_{1p}X_p}+\ e^{\beta_{21}+\beta_{01}X_1+...+\beta_{2p}X_p}} \quad (7)$$

where

$P(Y=0|X); P(Y=1|X); P(Y=2|X)$ = vectors of estimated probabilities;
$Y$ = dependent variable;
$\beta$ = vector of logistic regression coefficients;

$X = (X_1, \ldots, X_p)$ independent variables;
$p = 1, \ldots, n$.

The specification of the two econometric models, with indication of the multiple regression equation and identification of all operationalized variables, as well as the random disturbance term, is defined as follows:

$$\text{Model 1}: \text{Logit}\left(\hat{\pi}_p\right) = \beta_0 + \beta_1 X_{1p} + \beta_2 X_{2p} + \beta_3 X_{3p} + \beta_4 X_{4p} + \varepsilon_p \tag{8}$$

where

$\hat{\pi}$ = dependent variable QoL;
$X_1$ = independent variable SE;
$X_2$ = independent variable AE;
$X_3$ = independent variable CE;
$X_4$ = independent variable SIZE;
$\varepsilon_p$ = error (other factors/unobservable characteristics);
  with $p = 1, \ldots, 23$.

$$\text{Model 2}: \text{Logit}\left(\hat{\pi}_p\right) = \beta_0 + \beta_1 X_{1p} + \beta_2 X_{2p} + \varepsilon_p \tag{9}$$

where

$\hat{\pi}$ = dependent variable QoL;
$X_1$ = independent variable PS;
$X_2$ = independent variable SIZE;
$\varepsilon_p$ = error (other factors/unobservable characteristics);
with $p = 1, \ldots, 23$.

To contrast these results, a probit regression was also performed. Logit and probit regressions are similar because each returns sigmoid probabilities that sum to one over all alternatives; however, probit offers a potential advantage over logit in that the probit error specification allows correlations between the errors [93]; that is, for the logit models, the errors are assumed to follow the standard logistic distribution and for the probit the errors are assumed to follow a normal distribution [94].

## 6. Presentation and Discussion of the Results

### 6.1. First-Stage Results: DEA

A DEA was used to estimate the efficiency scores of the pro-sustainability activities of 23 public HEIs. In this phase, the 16 models presented in Table 7 were analysed and the means of the results (scores) are presented in Table 9 for each model (A–D), as well as the global average, variance, skewness and kurtosis. A radar chart is also presented in Figure 1, to facilitate visual inspection of the set of values obtained in the DEA analysis, by model.

As observed in Table 5, for the social efficiency activities (Model A), taking as a reference the average obtained per DMU, a homogeneous distribution is revealed, highlighting that none of them is below the median threshold of 50%. Four institutions recorded averages above 80% (UNL, UL, ISCTE and UAB); ten HEIs are between 60% and 80%; and nine obtained values between 50% and 60%. The global average of social efficiency is found to be 66%, with this being the highest average of the four models (A–D).

As for the average of Model B, the environmental efficiency activities, two HEIs stand out with averages above 80% (UAB and IPV); eleven are between 60% and 80%; two between 50% and 60%; and the remainder are below 50%. The global average of environmental efficiency is 60%.

**Table 9.** Average values (scores) by model: Data Envelopment Analysis (DEA).

| Higher Education Institutions (HEIs) | Mean Social Efficiency (Model A) | Mean Environmental Efficiency (Model B) | Mean Cultural Efficiency (Model C) | Mean Pro-sustainability Efficiency (Model D) |
|---|---|---|---|---|
| Universidade de Lisboa (UL) | 84.123 | 69.995 | 86.445 | 83.990 |
| Universidade do Porto (UP) | 58.078 | 42.735 | 52.465 | 57.060 |
| Universidade de Coimbra (UC) | 53.238 | 48.633 | 8.285 | 53.413 |
| Universidade Nova de Lisboa (UNL) | 85.495 | 73.028 | 85.048 | 85.218 |
| Instituto Politécnico do Porto (IPP) | 50.113 | 31.250 | 43.983 | 49.008 |
| Universidade do Minho UM | 57.108 | 24.535 | 5.873 | 52.658 |
| Universidade de Aveiro UA | 51.815 | 50.350 | 9.135 | 52.035 |
| Instituto Politécnico de Leiria IPL | 61.230 | 73.140 | 6.123 | 61.388 |
| Instituto Universitário de Lisboa ISCTE | 82.640 | 74.200 | 79.633 | 82.738 |
| Universidade do Algarve UAL | 58.115 | 79.733 | 19.965 | 63.125 |
| Universidade da Beira Interior UBI | 55.608 | 72.963 | 7.758 | 58.693 |
| Universidade de Évora UE | 54.408 | 55.340 | 7.195 | 54.890 |
| Universidade de Trás-os-Montes e Alto Douro UTAD | 59.273 | 43.340 | 4.923 | 58.505 |
| Universidade Aberta UAB | 82.555 | 96.385 | 98.950 | 84.770 |
| Instituto Politécnico de Viseu IPV | 76.610 | 89.285 | 9.408 | 78.788 |
| Instituto Politécnico do Cávado e Ave IPCA | 64.783 | 25.243 | 5.965 | 58.460 |
| Instituto Politécnico de Viana do Castelo IPVC | 69.043 | 45.578 | 25.933 | 67.625 |
| Instituto Politécnico de Castelo Branco IPCB | 71.468 | 69.305 | 11.268 | 69.258 |
| Instituto Politécnico de Santarém IPS | 66.348 | 60.045 | 17.495 | 67.138 |
| Universidade dos Açores UAC | 67.823 | 33.245 | 7.860 | 63.468 |
| Universidade da Madeira UMA | 68.913 | 77.718 | 6.670 | 71.020 |
| Instituto Politécnico de Portalegre IPPortal | 62.593 | 70.320 | 5.800 | 65.663 |
| Escola Superior de Enfermagem de Lisboa ESEL | 78.318 | 62.860 | 76.338 | 78.215 |
| Mean By model | 66.074 | 59.532 | 29.675 | 65.962 |
| Variance | 123.576 | 405.858 | 1055.814 | 131.997 |
| Skewness | 0.414 | −0.202 | 1.128 | 0.432 |
| Kurtosis | −1.053 | −0.812 | −0.379 | −1.039 |

Source: Own elaboration.

**Figure 1.** Distribution of the average values (scores), by model, of the DEA. Source: Own elaboration.

Model C, measuring cultural efficiency, reveal a heterogeneous distribution: three HEIs with an average above 80% (UAB, UL and UNL); two with averages between 60% and 80%; and the remaining sixteen with averages under 50%. The global average of cultural efficiency is only 30%.

Concerning pro-sustainability efficiency, Model D, including simultaneously the social, environmental and cultural factors, four HEIs are found to be above 80% (UNL, UAB, UL and ISCTE); ten between 60% and 80%; eight between 50% and 60%; and only one is below this, but very close to 50%. The global average of pro-sustainability efficiency is 66%. Regarding skewness and kurtosis, all the values indicate a normal distribution.

The radar chart in Figure 1 shows that, indeed, the values oscillating most are those related to cultural efficiency, followed by environmental efficiency. Social efficiency reveals the least variability and comes closest to the line referring to pro-sustainability efficiency.

### 6.2. Second-Stage Results: Multinomial Logit and Probit Regression

The second-stage analysis investigates whether the variation in efficiency can influence the dimensions characterising regional QoL. To do so, two selected model specifications were considered (see Table 7) and analysed through estimation of a multinomial logistic regression.

The first step is to produce descriptive statistics of the variables studied for each model. The distribution of the average values was found to be homogenous. The correlational relation between the variables, kurtosis, asymmetry and VIF were also analysed and the results reveal that all the values are within normality (see Table 10), except for the "cultural efficiency" correlation with QoL, which exceed the value of 0.7.

**Table 10.** Descriptive statistics, correlations, kurtosis, asymmetry and VIF among the variables.

| Variables Model 1 | 1 | 2 | 3 | 4 | 5 |
|---|---|---|---|---|---|
| QoL | 1 | | | | |
| Social efficiency | 0.661 ** | 1 | | | |
| Environmental efficiency | 0.312 | 0.536 ** | 1 | | |
| Cultural efficiency | 0.969 ** | 0.698 ** | 0.341 | 1 | |
| Size | 0.423 * | −0.087 | −0.338 | 0.374 | 1 |
| Mean | 0.629 | 66.074 | 59.532 | 29.675 | 0.348 |
| Variance | 0.098 | 123.58 | 405.86 | 1055.8 | 0.237 |
| Asymmetry | 1.079 | 0.414 | −0.202 | 1.128 | 0.684 |
| Kurtosis | 0.481 | 0.481 | 0.481 | 0.481 | 0.481 |
| VIF | (a) | 2.83 | 1.667 | 3.237 | 1.896 |
| **Variables Model 2** | **1** | **2** | **3** | | |
| QoL | 1 | | | | |
| Pro-sustainability efficiency | 0.661 ** | 1 | | | |
| Size | 0.423 * | −0.131 | 1 | | |
| Mean | 0.629 | 65.962 | 0.348 | | |
| Variance | 0.098 | 132.0 | 0.237 | | |
| Asymmetry | 1.079 | 0.432 | 0.684 | | |
| Kurtosis | 0.481 | 0.481 | 0.481 | | |
| VIF | (a) | 1.017 | 1.017 | | |

** The correlation is significant at 0.01 (2 extremities). * The correlation is significant at 0.05 (2 extremities). (a) Dependent variable: QoL. Source: Own elaboration.

The probability of each of the "efficiency" variations (0—weak; 1—average; and 2—high) was estimated from the QoL variable (material living conditions + health + education + environment + leisure + safety). All the models were adjusted with Stata software. Table 7 presents the estimates of the coefficients and respective outputs of the program for each of the eight models estimated. All the models are statistically significant ($p < 0.05$), except for Model 2. Concerning the quality of adjustment, the test statistic and the significance of the chi-squared tests are presented, with the results indicating the models are suitably adjusted. Unlike the likelihood-ratio, Wald and similar testing procedures, the models need not be nested to compare the information criteria [95]. Therefore, two statistics were performed to calculate the two information criteria used to compare the models: Akaike's information criterion (BIC) and the Bayesian information criterion (AIC).

In general, given the two models, the one with the smaller AIC fits the data better than the one with the larger AIC, as does a smaller BIC, indicating a better-fitting model. Table 11 shows the significant models and correspondent values.

**Table 11.** Coefficients of Model 1's multinomial logit and probit, with and without the control variable.

| Multinomial Logit Models | | | | | | Probit Models | | | | |
|---|---|---|---|---|---|---|---|---|---|---|
| Logit Model 1a QoL [a] | | Coef. | Std. Err | z | P > \|z\| | Probit Model 1a QoL | Coef. | Std. Err | z | P > \|z\| |
| Average variation (Dependent variable = 1) | Social efficiency | −1.740 | 1.049 | −1.66 | 0.097 * | Social efficiency | −1.163 | 0.585 | −1.99 | 0.047 ** |
| | Environmental efficiency. | 1.267 | 0.953 | 1.33 | 0.184 | Environmental efficiency. | 1.000 | 0.521 | 1.92 | 0.055 * |
| | Cultural efficiency | 2.297 | 1.126 | 2.04 | 0.041 ** | Cultural efficiency | 1.563 | 0.602 | 2.60 | 0.009 ** |
| | Constant | −0.906 | 1.013 | −0.89 | 0.371 | Constant | −0.634 | 0.557 | −1.11 | 0.265 |
| High variation (Dependent variable = 2) | Social efficiency | −3.351 | 2.416 | −1.39 | 0.165 | | | | | |
| | Environmental efficiency. | 3.704 | 1.629 | 2.27 | 0.023 ** | | | | | |
| | Cultural efficiency | 4.519 | 2.534 | 1.78 | 0.074 * | | | | | |
| | Constant | −5.476 | 2.620 | −02.09 | 0.037 | | | | | |
| Number of obs = 23 | | | | | | Number of obs = 23 | | | | |
| LR chi2(6) = 18.51 | | | | | | LR chi2(3) = 8.12 | | | | |
| Log likelihood = −15.150 | | | | | | Log likelihood = −9.247 | | | | |
| Prob > chi2 = 0.005 | | | | | | Prob > chi2 = 0.044 | | | | |
| AIC = 46.301 | | | | | | AIC = 26.496 | | | | |
| BIC = 55.385 | | | | | | BIC = 31.038 | | | | |
| Logit Model 1b QoL [a] | | Coef. | Std. Err | z | P > \|z\| | Probit Model 1b QoL | Coef. | Std. Err | z | P > \|z\| |
| Average variation (Dependent variable = 1) | Social efficiency (SE) | −2.013 | 1.181 | −1.70 | 0.088 * | Social efficiency | −1.345 | 0.659 | −2.04 | 0.041 ** |
| | Environmental efficiency. | 1.171 | 0.976 | 1.20 | 0.230 | Environmental efficiency. | 0.894 | 0.540 | 1.65 | 1.953 |
| | Cultural efficiency | 2.753 | 1.450 | 1.90 | 0.058 * | Cultural efficiency | 1.835 | 0.758 | 2.42 | 0.016 ** |
| | Size | −0.981 | 1.758 | −0.56 | 0.577 | Size | −0.646 | 0.985 | −0.66 | 0.512 |
| | Constant | −0.484 | 1.233 | −0.39 | 0.694 | Constant | −0.299 | 0.740 | −0.40 | 0.686 |
| High variation (Dependent variable = 2) | Social efficiency | −3.817 | 2.705 | −1.41 | 0.158 | | | | | |
| | Environmental efficiency. | 3.413 | 1.676 | 2.04 | 0.042 ** | | | | | |
| | Cultural efficiency | 5.313 | 2.987 | 1.78 | 0.075 * | | | | | |
| | Size | −1.903 | 2.529 | −0.75 | 0.452 | | | | | |
| | Constant | −4.660 | 2.783 | −1.67 | 0.094 | | | | | |
| Number of obs = 23 | | | | | | Number of obs = 23 | | | | |
| LR chi2(8) = 19.13 | | | | | | LR chi2(4) = 8.42 | | | | |
| Log likelihood = −14.843 | | | | | | Log likelihood = −9.027 | | | | |
| Prob > chi2 = 0.014 | | | | | | Prob > chi2 = 0.077 | | | | |
| AIC = 49.687 | | | | | | AIC = 26.037 | | | | |
| BIC = 61.042 | | | | | | BIC = 31.713 | | | | |

[a] The category of reference is: Weak variation (reference level). Level of significance * = $p < 0.100$; ** = $p < 0.050$.

Observation of Table 7 reveals that all the dimensions (social efficiency, environmental efficiency and cultural efficiency) have a significant effect on both models (Logit Model 1a: social efficiency: $p = 0.088$; environmental efficiency: $p = 0.023$; cultural efficiency: (average variation) $p = 0.041$, (hight variation) $p = 0.074$; Probit Model 1a: social efficiency: $p = 0.047$; environmental efficiency: $p = 0.055$; cultural efficiency: $p = 0.009$). When applying the control variable "Size" in Logit Model 1b, the introduction of this variable improves the significance of the "social efficiency" in the average variation ($p = 0.88$). Regarding environmental and cultural efficiency, when the control variable is introduced, the significance decreases. Probit Model 1b was insignificant.

As for AIC and BIC, when adding the control variable, the result is found to change slightly, but without much relevance. However, concerning the Multinomial Logit Models and Probit Models, this difference is greater, with the first fitting the data better.

These results indicate that all the dimensions are associated with increased levels of QoL, and all the dimensions are also influenced by the HEI's size. However, social efficiency is negatively related to intermediate levels of QoL, indicating that, probably, what is done within the HEI at a social level is not enough to have positive effects on the region's QoL. On the other hand, increasing the number of students makes this effect more negative. Regarding the effect of "size" on the influence of environmental and cultural efficiency on QoL, this may be lower if the number of students increases, since the control variable

decreases the significance of those relationships. In order to highlight the importance of social efficiency, environmental efficiency, and cultural efficiency, and the liaison with the size of the HEIs (control variable), Table 12 summarizes the statistically significant results found in the logit model, with and without this control variable.

**Table 12.** Coefficient and *p*-values of the social efficiency and size variables.

| Variables with Significant Value | Coef. | | *p*-Value | |
|---|---|---|---|---|
| | Without Control Variable "Size" | With Control Variable "Size" | Without Control Varable "Size" | With Control Variable "Size" |
| Social efficiency (average variation) | −1.740 | −2.013 | 0.097 * | 0.088 * |
| Environmental efficiency (high variation) | 3.704 | 3.413 | 0.023 ** | 0.042 ** |
| Cultural efficiency (average variation) | 2.297 | 2.753 | 0.041 ** | 0.058 * |
| Cultural efficiency (high variation) | 4.519 | 5.313 | 0.074 * | 0.075 * |

Level of significance * = $p < 0.100$; ** = $p < 0.050$. Source: Own elaboration.

### 6.3. Discussion

According to the results obtained from applying the DEA method and observation of Table 4, a pattern worthy of note is detected; i.e., HEIs with better pro-sustainability efficiency, especially in the social aspect, are located in the Greater Lisbon area. This result is not surprising and agrees with van Vught [49] and Lepori et al. [50] when stating that HEIs' positioning depends on the stock or resources available in the region. HEIs located in regions with greater resources (financial, logistic, physical, human capital, etc.) differentiate in being more efficient in transforming their resources and become more pro-active in anticipating changes and in developing the capacity to respond appropriately to the identified needs, as mentioned by Mazzarol and Soutar [51]. Indeed, regional asymmetries are greatly linked to both peripheral locations and the economic, social and institutional structures and dynamics of different regions [96]. More peripheral regions are usually expected to be less developed, as they are further from the main centres of decision-making, production and consumption [83]. Considering that if on one hand the HEIs must adapt to their surrounding population, and on the other that the population also ends up adapting to the existing educational supply, there is always a certain synergy between the characteristics of teaching, educational institutions and the local population/social context, as mentioned by [44]. So, it would be important to characterise the Portuguese higher education system and determine the presence of asymmetries between the various regions, assessing the different ways in which these institutions relate to their physical and social environment.

To respond to Q1, "Are HEIs efficient in transforming their inputs into pro-sustainability outputs?", two new insights are provided. Firstly, Portuguese HEIs manage to present intermediate levels of pro-sustainability efficiency, with social and environmental aspects showing the greatest efficiency. This result demonstrates the HEIs' concern about their social involvement in activities linked to the third mission [30], and each HEI's capacity to stimulate, for example, gender equality, direct (e.g., grants) and indirect (e.g., accommodation services, sport, psychological support, volunteerism, etc.) social support, combating academic drop-out and, more recently, support/encouragement for student mobility to peripheral regions. In the Portuguese case, as highlighted by some studies, for example in [34], social inequalities are very relevant when analysing the problem from a perspective more associated with income inequality or when focusing on the intersections and cumulative effects of various forms of educational, gender, territorial and ethnic inequality, etc. (e.g., [35,97]). The HEIs' contribution to their regions is through study grants awarded to needy students or those far from home, implementing activities to promote gender equality, both in terms of teaching and regarding the local population, and in implementing and extending social support to their students in particular, and to society in general.

Regarding the environmental contribution, although the results demonstrate that most HEIs manage to reach a reasonable level in transforming their inputs in environmental efficiency, there is certainly much work to be done. HEIs often have an important environmental concern on campus, but frequently the results do not extend to the surrounding regions, and if they do so, this is very localized and on a very small scale. These situations occur because regional entities do not have that concern about environmental sustainability or because there is not yet sufficient capital to develop the necessary infrastructure to accompany such activities. It is also necessary to develop greater environmental awareness through inter-generational education programmes. Regarding cultural efficiency, it was demonstrated that much remains to be done, principally in peripheral regions where resources and access to cultural goods are scarce or even non-existent.

Secondly, the HEIs presenting greater efficiency are located in the Greater Lisbon area. This may indicate that these institutions are well integrated in their region and present a differentiated, competitive orientation and positioning, being able to give greater prominence to activities directed to improving pro-sustainability efficiency, according to regional needs.

Therefore, the strategic path the HEIs follow, their structures and operations, are linked to the region wherein they operate [45]. This result also reflects the rapid and profound structural change in Portuguese society, resulting from the processes of social re-composition found over the last three decades, which underlined the country's regional asymmetries [35]. These authors mention the continued existence of inequalities, above all, in essentially rural regions more distant from major urban centres and their surrounding areas of influence, particularly in the regions of Alentejo, the Centre and the Autonomous Region of the Azores, with Greater Lisbon presenting values that tend to position this region in a more favourable wider context.

Regarding the second analysis, and to answer Q2, "What is the role of this efficiency as a predictor of regional QoL?", the results underline that the HEIs' pro-sustainability efficiency has a positive influence on the region's QoL, through environmental and cultural efficiency, but also reinforce the importance of the HEI's size, in terms of student numbers, as a component strengthening the significant effect of those dimensions. If the HEIs have more students, especially with regard to environmental and cultural efficiency, it can lead to a lower QoL in the region, which is justified by the fact that many times the agglomeration of students in a certain region can destabilize the lives of those that inhabit in that region, for example, with more noise, more garbage on the streets, more confusion, less security, etc. As mentioned by Goddart [9], HEIs are not just situated in places, they belong to their regions, as they interact with them in a diversity of ways. Therefore, the HEIs' pro-sustainability interaction with their regions of influence can take place in various ways, namely, through the students and staff who live in the region; activities of a social, environmental and cultural nature developed on and off campus; ethical social services, showing civic responsibility, provided to the community; and the creation of sustainable, ecological infrastructure on and off campus, etc. The whole dynamics should be ensured, considering the needs of both the HEI and its surrounding region, contributing to regions' attractiveness and sustainable development, and to inducing positive externalities with regard to regional QoL [79].

These results can be extrapolated to other regional realities, namely, in the European space, where there are national networks of public HEIs, aiming to promote territorial cohesion and social mobility through education, research and development, qualification, lifelong learning and, obviously, positively influencing the QoL of the regions.

## 7. Conclusions, Limitations, Research Agenda and Implications

This study assesses HEIs' pro-sustainability efficiency, considering the social, environmental and cultural factors, examining how their efficiency can influence regional QoL. The study uses a two-step methodology. In the first step, a standard DEA approach was used to estimate the efficiency scores of 23 Portuguese public HEIs; and in the second step,

a multivariate logit regression assessed the role played by the HEIs' pro-sustainability efficiency in the regional QoL.

The main findings reveal that HEIs located in the Lisbon region have a higher level of pro-sustainability efficiency, although that efficiency is more significant and positive in environmental and cultural factors. Regarding the contribution of the HEIs' pro-sustainability efficiency to the region's QoL, through the three dimensions of efficiency, the institution's size, in terms of student numbers, is shown to be a control variable contributing to the level of interaction between efficiency and regional QoL. In this analysis, the environmental component of efficiency was found to contribute most to regional QoL.

The article is innovative and contributes to the literature on HEIs' pro-sustainability efficiency in two ways: firstly, it maps the most efficient HEIs by collecting the key indicators (inputs and outputs) based on studies of HEIs' impact on their region and from data from a field study, in order to analyse the pro-sustainability efficiency (social, environmental and cultural), through constructing models that are estimated using the DEA method; and secondly, it analyses whether their efficiency influences the regional QoL through specification of logit and probit multivariate models, using the HEIs' efficiency scores as explanatory factors of regional QoL.

There are several limitations that must be underlined. Firstly, it is pointed out that only Portuguese HEIs were included, and so comparisons cannot be made with other international HEIs. However, significant and elucidative results were obtained for the Portuguese case, and the study can be replicated in other international higher education systems. Secondly, the limited number of HEIs under analysis, despite being justified by the unavailability of complete data regarding a greater number of institutions that take part in the scientific and technological system in Portugal. Nevertheless, the main public HEIs were included in the study. Thirdly, the was difficulty in gathering data at the NUTS III level, and especially concerning HEIs. Therefore, a suggestion for the future is to extend the population under study, including new samples of HEIs in other countries, for ensuring a higher number of DMUs and possibly prevent some potential bias present in reduced dimension samples. Fourthly, the fact is that the benchmarking exercise of the DEA analysis considers, by default, the best reference included in the DMU group. Fourthly, it can also be mentioned as another limitation the fact of using a limited set of indicators selected from the literature review. However, a large number of previous studies was reviewed, and the indicators found were tested and scrutinized in the scope of a field study undertaken with experts on higher education. Fifthly, the fact that there was no bootstrapping analysis in the deterministic DEA approach implemented to carry out the study may be an issue. Despite the various attempts made, the necessary convergence of the estimated parameters in the bootstrapping simulation was not ensured. This may be related to the reduced number of DMUs under analysis, already mentioned as a limitation of the empirical approach. Sixthly, a "static" view is presented here, since it was considered only for one year, which is why it is suggested, as an example of a research endeavour to be prosecuted, the future development of longitudinal studies.

Thus, in the light of the empirical evidence now obtained, it is necessary to pursue a future research agenda that includes longitudinal cross-country studies on the influence of the efficiency of HEIs on the regional QoL, to contrast the previous period and the period after the outbreak of the COVID$-$19 pandemic crisis, considering the different dimensions of the QoL, as recommended in the world reference initiative: "OECD Better Life Index". Additionally, it is suggested to continue the present study, through the development of a composite index that measures the efficiency of HEIs with pro-sustainability orientation, so that this index can be considered in financing decisions, both public and private, of this type of institutions.

The implications of the current study can be seen in two ways: firstly, through the type of association made, which strengthens knowledge about HEIs' influence on their regions, synthesizing at the same time the change in HEIs' social, environmental and cultural role, considering the population's QoL. Secondly, HEIs can reinforce the institutional orientation

of pro-sustainability management, and the study provides new lines for public policies devoted to strengthening HEIs' role in the necessary stimulation of more and better social and cultural activities, with environmental awareness, as levers of regional QoL.

Regional disparities are also connected to peripheral locations and to the economic, social, cultural and environmental structures and dynamics of the different regions. In this line of thought and argument, it is fundamental to consider the HEIs' history and location when making critical decisions on financing teaching, research and knowledge and technology transfer activities carried out by the HEIs, with a proven influence on regional QoL, and thereby emphasize the social, cultural and environmental components of efficiency required of these institutions, which are determinant for the education and absorption of sustainability values at the regional level.

**Author Contributions:** E.d.M.P., J.L. and H.A. contributed to the design and implementation of the research, to the analysis of the results and to the writing and revision of the manuscript. All authors have read and agreed to the published version of the manuscript.

**Funding:** This work was supported by the FCT—Fundação para a Ciência e a Tecnologia, in the scope of the research activities developed at the PTDC/EGE-OGE/29926/2017.

**Institutional Review Board Statement:** Not applicable.

**Informed Consent Statement:** Not applicable.

**Data Availability Statement:** The data presented in this study are available on request from the corresponding author. The data are not publicly available due to privacy of data sources.

**Acknowledgments:** The authors acknowledge the highly valuable comments and suggestions provided by the editors and reviewers, which contributed to the improvement in the clarity, focus, contribution and scientific soundness of the current study.

**Conflicts of Interest:** The authors declare there are no conflicts of interest.

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
