# Peer review of "HEI Efficiency and Quality of Life: Seeding the Pro-Sustainability Efficiency"

_sustainability, doi:10.3390/su13020514_

Round 1

Reviewer 1 Report

The work "HEIs Efficiency and Quality of Life: Seeding the pro-sustainability efficiency" has strengths and some weaknesses. I comment on them for your consideration.
- What is the main research problem found that has led you to make this manuscript?
- And the starting hypotheses?
- Limitations of the methodology?
- A section should discuss how these results could be extrapolated or highlighted with other studies in different regions (near or far).
- Have you found a similar study carried out in another region?
- References could be updated. Consider the following:
https://doi.org/10.3390/su12166338
https://doi.org/10.1080/07294360.2020.1830038
https://doi.org/10.3390/su12198254

Good luck

Reviewer 2 Report

Congratulations for this very interesting and novel approach on HEI. I really enjoyed reading and reviewing the paper. I think that the paper is very well written and in accordance with the scope of this journal.

I think however that there are some minor aspects that could be still improved. For instance in my opinion the introduction is quite to long. Some of the aspects from the introduction could be reshaped in the lit review section. 

The literature review is fine and very well developed. So is the methodological design. I have also found another paper that is dealing with different indicators in HEI, but not on sustainable aspects: https://papers.ssrn.com/sol3/papers.cfm?abstract_id=2579935

Congratulations with the paper and best luck in attracting relevant citations.

Reviewer 3 Report

This paper has several methodological weaknesses that need to be addressed by authors.

Authors should revise their manuscript according to the following suggestions.

Table 1 is unclear as there are several acronyms unexplained. See, for instance, SB. In this table indicators are grouped into some macro classes, depending on their focus. However, authors do not discuss effectively the criteria adopted to group these indicators. Additionally, it is unclear why authors consider indicators that are not under the control of HEI, such as “access to broadband internet” together with indicator that are mostly controlled by HE such as “Environmental invention patents registered by HEIs”.

I agree with authors that when the number of input and output variables is high compared to the number of DMUs a strategy to reduce their number is necessary. However, using the arithmetical average in which the individual indicators are included with equal weight may not lead to an acceptable proxy input/output variable (see lines 349-353). Scholars have suggested several approaches to reduce the number of inputs and outputs. See, for instance

Cherchye, L.; Kuosmanen, T. Benchmarking Sustainable Development: A Synthetic Meta-Index Approach. In Understanding Human Well-Being; McGillivray, M., Clarke, M., Eds.; United Nations University Press: Tokyo, Japan, 139–168, 2006.

Daraio, C.; Simar, L. Advanced Robust and Nonparametric Methods in Efficiency Analysis: Methodology and Applications. Springer, New York (USA), 2007.

lo Storto, C. Ownership structure and the technical, cost, and revenue efficiency of Italian airports. Utilities Policy 2018, 50, 175–193.

Soleimani-damaneh, M.; Zarepisheh, M. Shannon’s entropy for combining the efficiency results of different

DEA models: Method and application. Expert Syst. Appl. 2009, 36, 5146–5150.

As I understand, authors estimated HEI efficiencies by implementing a extremely high number of DEA models. However, how DEA models have been implemented has not been discussed, i.e. input vs output orientation, returns to scale, radial measure of efficiency vs SBM efficiency, etc. Please, motivate the choices relative to DEA implementation.

Lines 386-390: authors utilize arithmetic mean to aggregate results. However, this approach to reduce data does not allow have a Pareto-optimal measurement.

The econometric equation (e.g., Logit model) is not displayed. Why choosing two Logit models?

DEA provides deterministic measurements of efficiency. Thus, bootstrapped DEA should be employed to account for statistical bias. See

Simar, L.; Wilson, P.W. Estimation and inference in two-stage, semi-parametric models of productive efficiency. J. Econ. 2007, 136, 31–64.

Round 2

Reviewer 1 Report

Dear Authors,

Thank you for the improvement of the manuscript and consider the contributions of the reviewers.

Author Response

Dear Editor-in-Chief, Prof. Dr Marc A. Rosen

Dear Invited Editors of the Special Issue, Prof. Dr Daniel Schiller and Dr Verena Radinger-Peer

Firstly, we would like to acknowledge all the reviewers for the constructive feedback and suggestions on the previous version of the manuscript.

Secondly, we are very pleased to have had the opportunity to revise and resubmit the paper. Considering the answers to the questions raised, we provide a global overview of what was changed according to the review proposals and constructive recommendations provided by the reviewers.

Yours faithfully

The Authors

#

Section

Current Page(s)

Reviewer 1 Questions/Comments

Authors’ Answers

Thank you for the improvement of the manuscript and consider the contributions of the reviewers.

We acknowledge the reviewer’s comments for increasing the focus, clarity and scientific soundness of the manuscript. This is also addressed by the authors, in the following final note:

Acknowledgements: The authors acknowledge the highly valuable comments and suggestions provided by the editors and reviewers, which contributed to the improvement in the clarity, focus, contribution and scientific soundness of the current study.

Reviewer 3 Report

The authors have addressed almost all my concerns and have done an appreciable effort to revise their paper.

Please, consider these limitations of the study:

  • The deterministic DEA approach implemented to conduct the study, not performing bootstrapping
  • The “static” view because one year only was considered in the study

Please,  discuss further streams of research.

Author Response

Dear Editor-in-Chief, Prof. Dr Marc A. Rosen

Dear Invited Editors of the Special Issue, Prof. Dr Daniel Schiller and Dr Verena Radinger-Peer

Firstly, we would like to acknowledge all the reviewers for the constructive feedback and suggestions on the previous version of the manuscript.

Secondly, we are very pleased to have had the opportunity to revise and resubmit the paper. Considering the answers to the questions raised, we provide a global overview of what was changed according to the review proposals and constructive recommendations provided by the reviewers.

Yours faithfully

The Authors

#

Section

Current Page(s)

Reviewer 3 Questions/Comments

Authors’ Answers

Please, consider these limitations of the study:

The deterministic DEA approach implemented to conduct the study, not performing bootstrapping

The “static” view because one year only was considered in the study.

Please, discuss further streams of research.

We acknowledge the reviewer’s comments.

Considering the reviewer’s comment, the following sentences were added (see p. 23, line 730):

Fifthly, the fact that there was no bootstrapping analysis in the deterministic DEA approach implemented to carry out the study. Despite the various attempts made, the necessary convergence of the estimated parameters in the bootstrapping simulation was not ensured. This may be related to the reduced number of DMU under analysis, already mentioned as a limitation of the empirical approach. Sixthly, the “static” view here presented, since it was considered only one year, that is why it is suggested, as an example of research endeavour to be prosecuted, the future development of longitudinal studies.

Considering the reviewer’s comment, the following paragraph was inserted (see p. 24, line 737):

Thus, in the light of the empirical evidence now obtained, it is necessary to pursue a future research agenda that includes longitudinal cross-country studies on the influence of the efficiency of HEIs on the regional QoL, to contrast the previous period and the period after the outbreak of the COVID-19 pandemic crisis, considering the different dimensions of the QoL, as recommended in the world reference initiative: ‘OECD Better Life Index’. Additionally, it is suggested to continue the present study, through the development of a composite index that measures the efficiency of HEIs with pro-sustainability orientation, so that this index can be considered in financing decisions, both public and private, of this type of institutions.